# NIRAVARI: A Parsimonious Bio-Decisional Model for Assessing the Sustainability and Vulnerability of Rainfed or Groundwater-Irrigated Farming Systems in Indian Agriculture

Jacques-Eric Bergez [1,*] , Mariem Baccar [1], Muddu Sekhar [2,3] and Laurent Ruiz [3,4,5]

1    AGIR, INRAE, Université de Toulouse, F-31320 Castanet-Tolosan, France
2    Civil Engineering Department, Indian Institute of Science, Bangalore 560012, India
3    Indo-French Cell for Water Sciences, ICWaR, Indian Institute of Science, Bangalore 560012, India
4    UMR SAS, INRAE, Institut Agro, F-35000 Rennes, France
5    UMR GET, Université de Toulouse, CNRS, IRD, UPS, CNES, F-31400 Toulouse, France
*    Correspondence: jacques-eric.bergez@inrae.fr

**Abstract:** Groundwater irrigation is essential to sustain food production, and aquifer depletion represents a major sustainability challenge for humanity. There is a need for adequate modelling tools to assess the impacts of farming practices on groundwater resources with policy-makers and farmers in different contexts, especially in the case of smallholder farms in the tropics. We introduce the NIRAVARI model, which was designed to represent the Indian farming and water resource context. NIRAVARI is a parsimonious model integrating biophysical and decisional processes dealing with the farming system and the water table processes. A specific focus is given to how to irrigate with multiple water resources. Its formalisms include equations from well-tested published models for soil moisture and plant water stress simulations. The programming and graphic user interface is based on Excel VBA. We illustrate the ability of NIRAVARI to simulate a broad range of farmer adaptation strategies using four scenarios of cropping systems and water resources policies, and therefore, its interest for participatory scenario design and assessment with stakeholders.

**Keywords:** irrigation strategy; modelling; climate change; India; farming systems

## 1. Introduction

Groundwater depletion represents a major sustainability challenge for humanity in the 21st Century, given that groundwater is overexploited worldwide, particularly for agriculture [1]. About 38% of global consumptive irrigation water demand is met by groundwater [2]. With being a decade away from Sustainable Development Goals, the need for concrete policies for groundwater management has become urgent, and all water authorities call for methods to assess the effectiveness of policies. Furthermore, climate change is expected to intensify this threat, especially in regions where irrigation sustains agriculture and where population increases. Both influence the need for adaptation strategies, and for methodological approaches to identify and evaluate these same strategies. This is the case in India, where irrigation by groundwater is fully deployed [3].

The awareness that water resource management must account for interactions and feedback between biophysical processes determining movement of water and human behavior in a given socio-economic context has gained significant recognition among scientists in the past few years [4]. Different approaches exist to work on such interactions. Scenario modelling and evaluation is a standard way to explore adaptation strategies in the water resource domain in the context of climate change [5]. These methods are rooted in the more classical Story and Simulation approach [6]. They are often associated with participatory approaches and, in recent decades, several participatory stakeholder frameworks have been designed and implemented in projects that address adaptation strategies for groundwater management [7].

Simulating decisions on the farm and the technical operations of the system is a good way to test management practices, as it integrates the dynamics of resource sharing and constraints of the farm depending on the weather and the crop's phenological stage [8]. Linking soil-crop models and decision-making models is an appropriate approach for analyzing the adaptation of practices to face new agricultural challenges, such as climate change [9]. However, in order to use modelling and simulation as a tool to provide new insights on the impacts of different water policies on farmers and on groundwater resources, there is a need for simple models which can be utilized to assess scenarios with policy-makers, extensionists and farmers [10]. However, given the large diversity of farming systems in the world, there is no "one fits all" model which could adequately represent all types of agriculture. In particular, few models can represent smallholding farms, which account for a large proportion of agriculture in Africa and South Asia, and especially in India [11]. Such a model needs to be parsimonious [12], integrate farmers' decision-making on crops and crop management [13], allow integrating rainfed crops and represent the effect of conjunctive use of surface water and groundwater on the resource and different irrigation techniques [14].

One can find different types of numerical models to manage irrigation for Indian agriculture. Some are based on testing the adequacy of different crop models for the specific conditions in India (see for example Aquacrop and DSSAT-CERES in [15] or [16]). Others are more concerned with more specific biophysical processes, such as salinity [17]. Others are based on linear programming and optimization processes [18], or on artificial intelligence with neural networks and remote sensing data [19] or even the Internet of Things and machine learning [20]. Actually, very few take into account the farmer's decision-making process as proposed by [21,22]. Ref. [23] represents an interesting attempt, but is mainly focused on channel irrigation that is not the only water resource for irrigation.

Our paper presents NIRAVARI, a simple and parsimonious biodecisional model, able to test policy schemes regarding pond, borewells and irrigation equipment. The paper mainly focuses on the farmer's decision-making part of the model, and the complexity of dealing with different water resources to irrigate their farms. The initial ideas of the model were initiated with some policy-makers during a workshop held in Bangalore in March 2019. It was then quite clear that a simple tool was necessary to test a large range of scenarios. Section 1 presents the Indian context, regarding farming system and irrigation management. Section 2 presents the model; i.e., the formalism and the equations. At the end of this section, we present the different scenarios simulated to demonstrate the ability of the model. Section 3 presents the model developed and we elaborate on the outputs of the simulated scenarios. In a final section, we discuss the potential and limits of the model to elaborate and test irrigation strategies at the farm level with stakeholders.

## 2. Materials and Methods

### 2.1. Specificities of Groundwater Irrigated Farms in India

The dramatic development of groundwater irrigation in the recent decades in India has increased the irrigated area, simultaneously increasing food production to sustain the demand of a growing population. However, this came at the cost of tremendous impacts on energy consumption [24], resource depletion [25] and pollution [26,27], which now calls for an urgent rationalization of groundwater use. However, solutions are not straightforward, and recent detailed studies conducted in a groundwater-irrigated region in southern India, belonging to the Kabini Critical Zone Observatory in Karnataka ([28]; SNO M-tropics, https://mtropics.obs-mip.fr/, accessed on 11 October 2022) have illustrated the complex interactions and feedback between water resource availability and cropping systems, documenting the complex technical functioning of a diversity of farming systems [29,30]. Most Indian farmers are smallholders (less than 1ha on average), and their land is composed of one or several independent pieces of land, called "jeminu", in Karnataka. The size of a jeminu can vary greatly (from 0.1 to several hectares), but typically are about half a hectare, which can be divided into several plots for growing different crops in rotation. When

jeminus are irrigated, they share the same water sources: a borewell and/or, more rarely, a farm pond. In hard rock aquifers, pump yields are small and often insufficient to fully satisfy the needs of water-intensive crops on large surfaces [25], and they vary in time with water table level as hydraulic conductivity decreases sharply with depth [31,32]. In addition, the energy required to operate submersible pumps is provided by the government at no cost but for a limited number of hours per day. Therefore, irrigated cropping systems are still largely dependent on rainfall, and farmers need to elaborate complex strategies involving the combination of different crops in the same jeminu, and allocate the adequate irrigation water to each of them to optimize their crop yield and economic returns. The system can become even more complex, as farm ponds, encouraged by local governments for securing alternate sources of irrigation and/or for enhancing the aquifer recharge [14], are increasingly present, adding further complexity to the system. Model-based decision support systems can therefore be of great help to farmers and decision makers in assessing a range of irrigation strategies.

*2.2. Overall Description of the System*

The modeled system (Figure 1) is composed of a jeminu (i.e., a farm) divided in *n* beles of the same size (i.e., plots). A jeminu can be rainfed or have a pond (water reserve) and/or a borewell equipped with a submersible pump. The pond can either be used to irrigate the bele or to recharge the groundwater table. The pond is represented as an inversed truncated pyramid. The surface of the pond can be removed from the productive area of the jeminu or not. This is an important feature for very small farms. The pump is used to irrigate the beles, but may also be used to refill the pond. The pump withdraws water from the aquifer and its flow rate depends on the pump characteristics and on the groundwater table depth. On each bele, a different crop can be grown. The crop growth depends on the duration since sowing. Crop management is described in a descriptive manner, i.e., date and action (sowing or harvest). Only irrigation practices follow a more complex decision-making process formalism. Different types of irrigation techniques can be used (e.g., drip, sprinkler or furrow) which differ by the amount of water they provide when irrigation is triggered. The model is run at a daily time step. Exogenous variables are the climatic values: rainfall ($R(t)$, mm) and potential evapotranspiration ($E_0(t)$, mm).

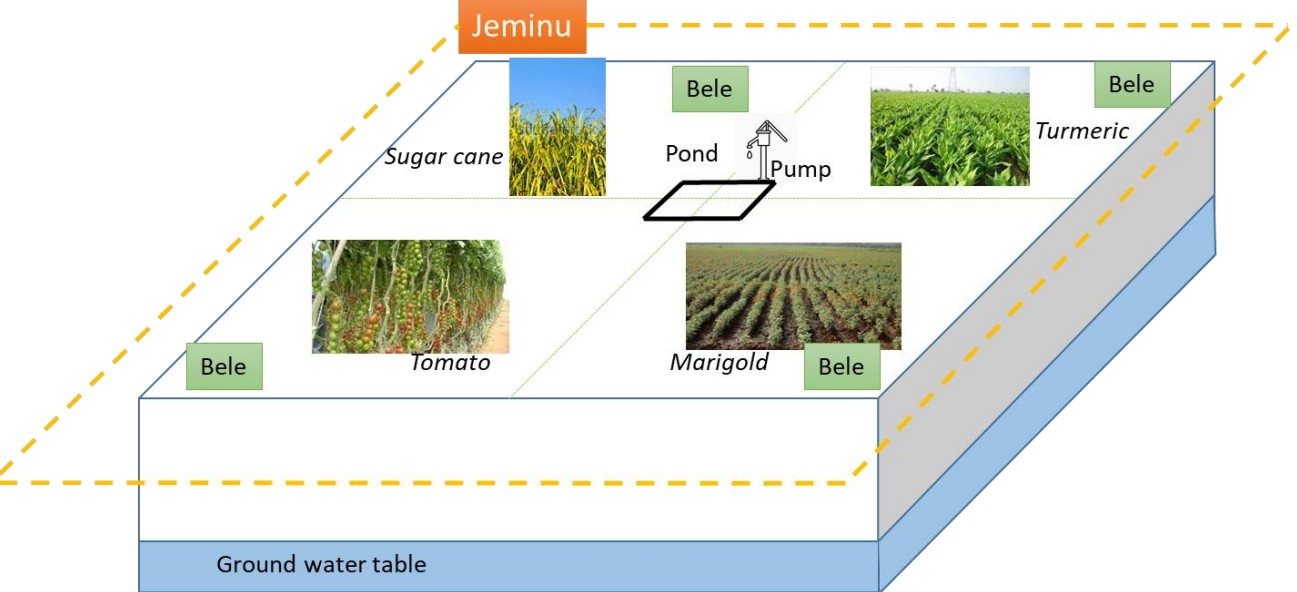

**Figure 1.** Schematic representation of a jeminu.

In the following, we describe the decision-making aspect of the model. The main biophysical equations that are more "classical" are given in Appendix A. Crop and water budget processes are based on FAO56 [33]. This decision-making model is based on farmers' interviews carried out in 2015 [29]. This survey was aimed at understanding farmers' farms structure and farmers' assets, and specific attention was given to the irrigation system: whether they are irrigating; whether they are using a pond and/or borewells; irrigation costs; how they pay for the irrigation systems and other equipment. A total of 684 farmers were interviewed with the help of local translators from September 2014 to March 2015, which represents 12.5% of farms in the watershed. The survey consisted of a face-to-face interview lasting 2–3 h. The survey was divided into three parts. The first part focused on household characteristics, farm structure, assets, partnerships and farm objectives. In the second part, we asked farmers about their performances and practices over the past two years. In the last part, in-depth questions were asked about irrigation, borewells and rainfall. Since no yearly records were kept by farmers, information about historical management went no further than the past two years. To develop the decision-making model, we used the approach developed by [34].

*2.3. Decision-Making Processes*

2.3.1. Management Processes at the Bele Level

On each bele, there is a decision model that allows: (i) to sow the crop; (ii) to harvest the crop; (iii) to irrigate the crop.

Sowing/Harvesting/Crop Succession

Sowing, harvesting and crop succession are based on a prescriptive model; i.e., the user gives the crop succession and the date to sow. Harvest time is automatically calculated by the model. This corresponds to the time it takes for the crop to reach maturity. For example: "*Beetroot is sown on the 1st of April and harvested on the 1st of July. It is followed by an Onion, sown on the 1st of August and harvested on the 4th of November. A fallow period follows during the Summer season.*"

Irrigation Model at the Bele Level

The irrigation campaign for each bele depends on the crop grown on the bele (1). The campaign is bordered by a starting crop age ($I_1$) and an ending crop age ($I_2$). Between these two thresholds, irrigation can occur if the crop is stressed. The stress is defined as the ratio of the actual evapotranspiration (*AET*) and the maximum evapotranspiration (*MET*):

$$if \left[ (I_1 < A(t) < I_2) \; and \; (\frac{AET(b, t-1)}{MET(b, t-1)} < I_3) \right] then \; I(t) = TRUE \tag{1}$$

The irrigation amount depends on the irrigation technique chosen at the bele level, and on the available water for irrigation either pumped from the ground water table or coming from the pond. Three irrigation technique are available: furrow, drip and sprinkler; but others may be added.

2.3.2. Management Processes at the Jeminu Level

Irrigation Model at the Jeminu Level

Competition between the irrigation required by each bele and the availability of water for the jeminu is managed in the irrigation model at the jeminu level. This is an adaptation of the algorithm developed in [35]. The general principles are as follows (Figure 2):

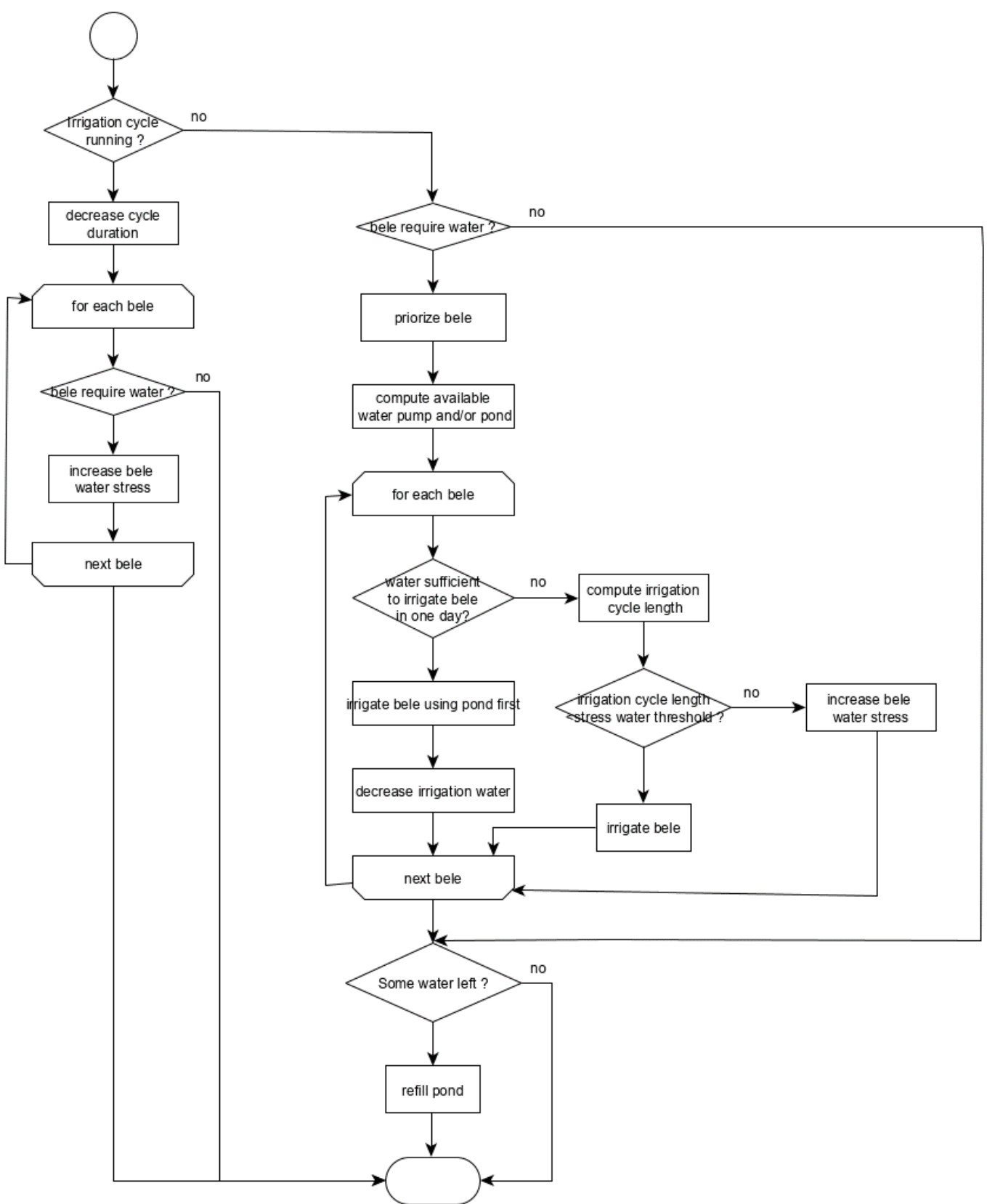

**Figure 2.** The general algorithm of the irrigation decision-making model at the jeminu level.

- Each day, the model checks if an irrigation cycle is running. If yes, the duration of the cycle is decreased by one, and the crops on the beles that would have needed irrigation are stressed by one more point.
- If there is no irrigation cycle running, then the model checks if the different beles need irrigation. Irrigation triggering once irrigation campaign has started is based on Equation (1).
- Priority between beles requiring irrigation is computed first (see below) and irrigation is then performed one after the other in decreasing priority.
- Irrigation water may come either from the pond or the groundwater table (through the pump).
- When irrigation is requested, if the pond can be used for irrigation, its water volume is used first to irrigate.
- For a bele needing irrigation, the amount provided by the irrigation cycle depends on the type of irrigation equipment (drip, sprinkler or furrow).
- The amount of irrigation water is given on the first day of the irrigation cycle on the bele.
- However, the amount to refill the pond (if any) is given on the last day of the irrigation cycle.
- If the amount required can be given in one day, the bele is irrigated and the remaining water can be used to irrigate another bele or to fill the pond, depending on the number of beles requiring irrigation. This is an optional possibility.
- If several days are required to irrigate, the irrigation cycle length is computed.
- If the irrigation cycle length may lead to crop failure (too large water stress), the bele is not irrigated and the next bele is tested for irrigation.
- If some water remains, it can be used to refill the pond.
- When an irrigation is performed, the pump is "blocked", meaning that it cannot be used for another purpose.

Irrigation Priority between Beles

If different beles require irrigation the same day, priority rules are needed if not enough water is available. To calculate irrigation priorities, we use a weighted average approach corresponding to a priority index, $\Re$, calculated for each bele (2). The priority bele is the bele with the smallest $\Re$. This approach makes it possible to integrate all the factors that impact the decision-making related to the management of irrigation. The choice of factor coefficients makes it possible to build different irrigation strategies.

$$\Re = a \cdot R_{crop} + b \cdot R_{stress} + c \cdot R_{age} + d \cdot R_{technic} \tag{2}$$

where $a$, $b$, $c$ and $d$ are the weight coefficients.

- $R_{crop}$ represents the crop priority.
- $R_{stress}$ represents the water stress priority factor. This factor is linked to the cumulative stress of the crop (number of days without water supply). For this factor, two strategies are possible: (i) favor the most stressed crops to prevent them from failing, or (ii) favor the least stressed crops, because the most stressed have already lost their yield potential.
- $R_{age}$ represents the crop cycle priority factor. This factor is linked to the relative (normalized) state of achievement of the culture cycle. For this factor, two strategies are possible: (i) favor crops close to the end of the cycle to ensure harvest, or (ii) favor crops at the start of the cycle, to ensure their successful development during the first phases of the cycle.
- $R_{technic}$ represents the irrigation amount priority factor. This factor is linked to irrigation technique (drip, sprinkler and furrow). The idea is to classify beles by their irrigation technique; from the techniques requiring the least water to the techniques requiring the most water. Two strategies are possible: (i) favor the irrigation technique with the

larger amount of water to provide; or (ii) favor the irrigation technique with the less amount of water to provide.

$\Re$ is calculated for each bele requiring irrigation. In case of ex aequo, a final factor is used, $R_{bele}$, the bele priority factor. The advantage of this formalism is that we separate the strategy from the code and give the user the chance to choose his own irrigation strategy. An example of this algorithm is given in Table 1. The first four lines of the table {1–4} provide the priority given to the four priority factors. $R_{crop}$ is priority one, $R_{stress}$ priority 2, $R_{age}$ priority 3 and $R_{technic}$ is not considered. For $R_{stress}$ and $R_{age}$, the option of how to consider stress and age is given, respectively. For $R_{stress}$, we favor the more stressed crop first; while for $R_{age}$, we favor first the more aged crop.

**Table 1.** An example of the algorithm to determine the priority between the different beles to be irrigated. In this case, the bele irrigated is bele 1 as it has the lowest $\Re$. See text for details.

| | | Priorities | Option | |
|---|---|---|---|---|
| 1 | $R_{crop}$: crop priority | 1 | | |
| 2 | $R_{stress}$: stress priority | 2 | favor more stressed crop | |
| 3 | $R_{age}$: crop cycle priority | 3 | favor aged crop | |
| 4 | $R_{technic}$: irrigation technic priority | 0 | - | |
| | | Weights | | |
| 5 | $a$ | 0.167 | $=1/(1 + 2 + 3)$ | |
| 6 | $b$ | 0.333 | $=2/(1 + 2 + 3)$ | |
| 7 | $c$ | 0.500 | $3/(1 + 2 + 3)$ | |
| 8 | $d$ | 0.000 | $0/(1 + 2 + 3)$ | |
| | | Daily data | | |
| 9 | Bele | 1 | 2 | 3 |
| 10 | Crop | cucurma | beetroot | onion |
| 11 | crop priority | 1 | 2 | 3 |
| 12 | Irrigation | Drip | Furrow | Sprinkler |
| 13 | Irrigation amount (mm) | 15 | 50 | 25 |
| 14 | Crop length (d) | 240 | 90 | 90 |
| 15 | Crop age (d) | 230 | 30 | 2 |
| 16 | Crop water stress level | 10 | 6 | 5 |
| 17 | Number of days without irrigation | 5 | 5 | 4 |
| | | Priority calculation | | |
| 18 | $R_{crop}$ | 0.167 | 0.333 | 0.500 |
| 19 | $R_{stress}(j)$ | 0.50 | 0.17 | 0.20 |
| 20 | $R_{age}(j)$ | 0.04 | 0.67 | 0.98 |
| 21 | $R_{technic}$ | 0.3 | 1 | 0.5 |
| 22 | Overall priority | 0.22 | 0.44 | 0.64 |
| 23 | In case of equal $R_{bele}$, bele prioriy | 1 | 3 | 2 |

The next four lines {5–8} give the computation for the different weights (see (2)). For example, as $R_{crop}$ is priority 1 and that {priority($R_{crop}$) + priority($R_{stress}$) + priority($R_{age}$) + priority($R_{technic}$)} equals 6, then ($a$) equals 1/6 = 0.167.

Lines {9–17} give the status of the system on an example day to demonstrate the computation: there are three beles, with each a different crop {10}. Priority between these different crops is given on {11}. These three beles can be irrigated with a different irrigation equipment {12} providing a given amount of water {13}. Lines {14–17} give the status of the crop regarding their age and their water stress. From this information the different priorities are calculated {18–21}. $R_{crop}$ is calculated using {11} and the same algorithm used to calculate the weights (see above); $R_{stress}$ is calculated as {17}/{16}; $R_{age}$ is calculated as

1- {15}/{14}; and $R_{technic}$ is calculated as {13} divided by the value of the technic given the largest amount of water; here, it is a furrow with 50 mm. Line {22} calculates the final value of $\Re$ using (2). In the case of ex aequo, we use {23} to decide the bele to irrigate.

2.3.3. The Dynamic of the System

The general dynamic of the model is as follows:

1.　Initialization of the simulation:

    1.1.　Create the jeminu.
    1.2.　Create the different beles and initialize the bele (soil water amount).
    1.3.　Create a pump and a pond, if any.
    1.4.　Initialize the pond (water amount).
    1.5.　Create a dictionary of crops and a dictionary of irrigation practices.
    1.6.　Read the full climatic series.

2.　Daily simulation:

    2.1.　For each bele, manage the crop (check for sowing, harvesting and crop failure).
    2.1.　On the jeminu, manage the irrigation.

        2.1.1.　Check for beles that need irrigation.
        2.1.2.　Manage priorities between beles.
        2.1.3.　Provide irrigation if water is available.

    2.3.　Update crop water stress on the different beles.
    2.4.　Perform the water budget on the different objects (beles; pond, if any; and groundwater table)

3.　At the end of the simulation period:

    3.1.　Write an output file (dump memory).
    3.2.　Create graphs to analyze the scenario.
    3.3.　Clean the memory and end-up the simulation.

*2.4. Modeling Approach*

The model was developed under VBA Excel© using an object modeling structure [36]. Graphic user interface was based on an Excel worksheet facilitating data validation and error checking. Each simulation creates a new workbook with simulated data and explanatory graphs.

*2.5. Testing Scenarios*

In order to demonstrate the use of the model, we proposed a set of example scenarios:

1.　Sc0 (ref): The reference scenario corresponds to a rainfed farm with two plots (5000 m$^2$ each). For rainfed farms, crops are grown only in two seasons, Kharif and Rabi, which is a widely used crop practice in the area. For each season, the farmer grows the two major crops of the area, each crop in one plot: for Kharif, sorghum and sunflower, and for Rabi, maize and horse gram. Rainfed crops are drought resistant, but these crops fail at a stress of 0.2 or less over a 5-day period.

2.　Sc1: For Scenario 1, the farmer chooses to keep the same cropping system as in Sc0, but to install a pond that is filled by the runoff of rainwater on his watershed. This pond is lined and covered in order to avoid losses and to make the best use of the collected water, in protective irrigation. The surface of the pond is removed from the jeminu area. The irrigation is triggered when the crop is stressed to 0.4, to avoid its failure. For this scenario, furrow irrigation is used, the most common irrigation technique in the area. In case of competition for irrigation water, the farmer favors the most stressed crop closest to the end of its cycle in order to maximize the chance of a harvest.

3.　Sc2: For Scenario 2, the farmer intensifies his production system compared to Sc0, by installing an 80 m-depth borewell, and by developing cash crops on four plots. The farmer grows beetroot in Kharif, tomatoes in Rabi and watermelon in Summer. These

crops are quite common in the area. Crop failure occurs when a crop has a stress less than or equal to 0.45 over a 5-day period (Table 2). The irrigation system is a furrow, and it is activated when the crop is at 0.6 stress. In case of competition for irrigation water, the farmer favors the crop with the highest stress level in order to avoid losing production quality.

4.　Sc3: For Scenario 3, the farmer has the same cropping system and the same structural aspects as Sc2, but he decides to install a pond in addition. The functioning of this pond is different from that of Sc1. The pond in Sc3 is filled with water from the borewell in addition to the runoff, in order to store water on days when there is no irrigation demand, or if there is excess water after irrigation.

**Table 2.** Crop parameters regarding crop water stress (see A3 for details).

| Crop | p1 [0; 1] | p2 [mm] | p3 [Number of Days] | Scenario |
|---|---|---|---|---|
| Beetroot | 0.45 | 20 | 5 | 3 |
| Horse gram | 0.2 | 20 | 5 | 1, 2 |
| Maize | 0.2 | 20 | 5 | 1, 2 |
| Sorghum | 0.2 | 20 | 5 | 1, 2 |
| Sunflower | 0.2 | 20 | 5 | 1, 2 |
| Tomato | 0.45 | 20 | 5 | 3 |
| Watermelon | 0.45 | 20 | 5 | 3 |

For all the scenarios, the farm size assumed is 1 ha with a typical black soil. We used the 15-year Maddur climatic series (Latitude: 12°35′3.01″ N; Longitude: 77°02′41.64″ E) to run the simulation. The full parametrization of Sc0 is given in Appendix B.

## 3. Results

### 3.1. Model Graphic User Interface

When the user opens NIRAVARI, only three worksheets are initially visible: (i) a "read me" sheet explaining how to use the model; (ii) a Climate sheet, allowing the integration of the requested climatic data; and (iii) an Init sheet, allowing the user to parametrize the simulation. The Graphic User Interface allows the user to parametrize the simulation (Figure 3) with different validation processes (based on "data validation" from Excel). Information is given to the user whereby they have to fill in some data. Only cells in yellow can be modified. Once the parametrization is performed, the user clicks on the "Run" button, allowing for the simulation to run. When the simulation is over, a new workbook is opened with the computed data and some standard graphs, to then analyze the simulation.

A second button, "Load", allows the user to load a previous run to test a modification on a base scenario.

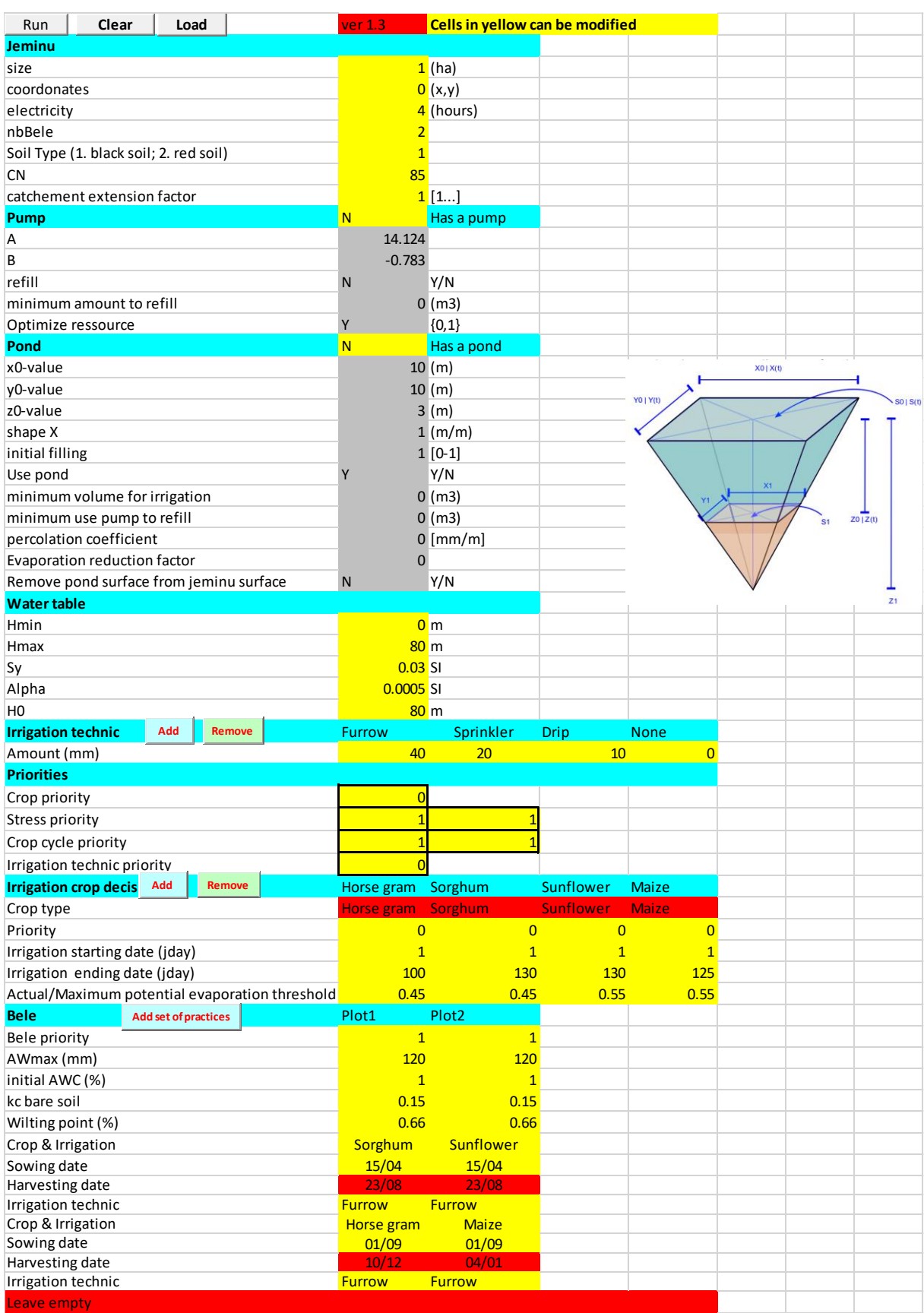

| Run | Clear | Load | | ver 1.3 | Cells in yellow can be modified | | | |
|---|---|---|---|---|---|---|---|---|
| **Jeminu** | | | | | | | | |
| size | | | | 1 | (ha) | | | |
| coordonates | | | | 0 | (x,y) | | | |
| electricity | | | | 4 | (hours) | | | |
| nbBele | | | | 2 | | | | |
| Soil Type (1. black soil; 2. red soil) | | | | 1 | | | | |
| CN | | | | 85 | | | | |
| catchement extension factor | | | | 1 | [1...] | | | |
| **Pump** | | | | N | Has a pump | | | |
| A | | | | 14.124 | | | | |
| B | | | | -0.783 | | | | |
| refill | | | | N | Y/N | | | |
| minimum amount to refill | | | | 0 | (m3) | | | |
| Optimize ressource | | | | Y | {0,1} | | | |
| **Pond** | | | | N | Has a pond | | | |
| x0-value | | | | 10 | (m) | | | |
| y0-value | | | | 10 | (m) | | | |
| z0-value | | | | 3 | (m) | | | |
| shape X | | | | 1 | (m/m) | | | |
| initial filling | | | | 1 | [0-1] | | | |
| Use pond | | | | Y | Y/N | | | |
| minimum volume for irrigation | | | | 0 | (m3) | | | |
| minimum use pump to refill | | | | 0 | (m3) | | | |
| percolation coefficient | | | | 0 | [mm/m] | | | |
| Evaporation reduction factor | | | | 0 | | | | |
| Remove pond surface from jeminu surface | | | | N | Y/N | | | |
| **Water table** | | | | | | | | |
| Hmin | | | | 0 | m | | | |
| Hmax | | | | 80 | m | | | |
| Sy | | | | 0.03 | SI | | | |
| Alpha | | | | 0.0005 | SI | | | |
| H0 | | | | 80 | m | | | |
| **Irrigation technic** | Add | Remove | | Furrow | Sprinkler | Drip | None | |
| Amount (mm) | | | | 40 | 20 | 10 | 0 | |
| **Priorities** | | | | | | | | |
| Crop priority | | | | 0 | | | | |
| Stress priority | | | | 1 | 1 | | | |
| Crop cycle priority | | | | 1 | 1 | | | |
| Irrigation technic priority | | | | 0 | | | | |
| **Irrigation crop decis** | Add | Remove | | Horse gram | Sorghum | Sunflower | Maize | |
| Crop type | | | | Horse gram | Sorghum | Sunflower | Maize | |
| Priority | | | | 0 | 0 | 0 | 0 | |
| Irrigation starting date (jday) | | | | 1 | 1 | 1 | 1 | |
| Irrigation ending date (jday) | | | | 100 | 130 | 130 | 125 | |
| Actual/Maximum potential evaporation threshold | | | | 0.45 | 0.45 | 0.55 | 0.55 | |
| **Bele** | Add set of practices | | | Plot1 | Plot2 | | | |
| Bele priority | | | | 1 | 1 | | | |
| AWmax (mm) | | | | 120 | 120 | | | |
| initial AWC (%) | | | | 1 | 1 | | | |
| kc bare soil | | | | 0.15 | 0.15 | | | |
| Wilting point (%) | | | | 0.66 | 0.66 | | | |
| Crop & Irrigation | | | | Sorghum | Sunflower | | | |
| Sowing date | | | | 15/04 | 15/04 | | | |
| Harvesting date | | | | 23/08 | 23/08 | | | |
| Irrigation technic | | | | Furrow | Furrow | | | |
| Crop & Irrigation | | | | Horse gram | Maize | | | |
| Sowing date | | | | 01/09 | 01/09 | | | |
| Harvesting date | | | | 10/12 | 04/01 | | | |
| Irrigation technic | | | | Furrow | Furrow | | | |
| **Leave empty** | | | | | | | | |

**Figure 3.** The upper screen of the graphic user interface. This Graphic User Interface allows for parametrizing of the simulation. Information and data validation are performed at this step.

### 3.2. Playing with the Model

The first indicator we observed was the level of crop failure over the 15-year climatic period (Table 3). Crop failure process is explained in A3. Growing using a rainfed system, (Sc0) shows a failure of 15% of the crops developed over the 15 years. Adding a pond in the jeminu (Sc1) improves the result (8% failure). This is due to the use of some irrigation to remove water stress to crops. However, it is impossible to avoid all failures, since the amount of water in the pond is not sufficient to irrigate all the crops in case of drought years. Moving to a pump and borewell system with more water-sensitive crops (but more expected economic return), (Sc2) shows a higher percentage of crop failure (29%). This high level of crop failure, despite irrigation possibilities, is due to two factors. The first is the use of more water-deficit-sensitive crops (Table 2). The second is due to the irrigation process itself. All beles cannot be irrigated on the same day, and the amount provided to the crops due to the irrigation technique (furrow) implies long irrigation cycles. Some beles are therefore not irrigated when needed, increasing the crop water stress leading to crop failure. This demonstrates the impact of the choice of water-intensive crops in spite of the addition of a bore well. If a buffer pond is added (Sc3), the situation improves with only 8% of crop failure.

**Table 3.** Percentage of crop failure over the 15-year climatic serie.

| Scenario | Number of Crops | Failure |
|----------|-----------------|---------|
| Sc0 | 60 | 15% |
| Sc1 | 60 | 8% |
| Sc2 | 180 | 29% |
| Sc3 | 180 | 8% |

The second indicator is an economic estimation (Figure 4). We used the net return indicator (*NR*) to compare the scenario estimated as (3):

$$NR = \sum_{i=1}^{n} [A_i \,/10,000 \cdot ((Y_i * w_i * p_i) - C_i \,] \tag{3}$$

where $A_i$: crop area (m$^2$), $Y_i$: crop potential yield (T·ha$^{-1}$), $w_i$: mean crop water stress; $p$: crop price (Rs·t$^{-1}$); $C_i$: crop costs (Rs·ha$^{-1}$). We carried out a survey with farmers in the region to estimate the production costs, the market price and the potential yield for each crop. In case of crop failure, which can occur at any time during the crop cycle, we estimate that the farmer loses all production costs.

Sc0 situation allows farmers to earn 48,600 Rs (median), with a consequent number of years of loss. Adding a pond for protective irrigation (Sc1) drastically reduces the years of losses and low net return. The system with irrigated crops with a borewell (Sc2) improves the median net return (336,000 Rs), but makes it very variable due to the crops' failure and high-value crops' sensitivity to water shortages and droughts. Adding a buffer pond (Sc3) not only improves the median net return (489,000 Rs), but also reduces variability.

The third indicator is based on the pond water budget (Figure 5). This figure is more complex as it integrates the different components used to compute the dynamic of the pond water budget. The dynamic of the volume of the pond is given in clear blue on the first y-axis (left-hand). Input variables for the pond are on the first y-axis: rain (red), refill from the borewell (yellow) and runoff from the watershed (gray). The output variables are given on the second y-axis (right hand) in inverse order, the zero value being at the top of the graph: pond over flooding (orange), pumped for irrigation (lilac) and evaporation (light red). Depending on the parameters used for simulation, some of these variables will be null, or will change during the simulation. Scenarios Sc1 and Sc3 show two different modes of using the pond. In Sc1, the pond contributes to storing runoff water for irrigation when crops are most stressed. The pond is filled from time to time and quite slowly (Figure 5A). In Sc3, the pond is used as a buffer in order to store the water available by the pump and

by the runoff. The pond is therefore filled quite often. It is used when the pump flow rate is not sufficient to provide the required irrigation needs.

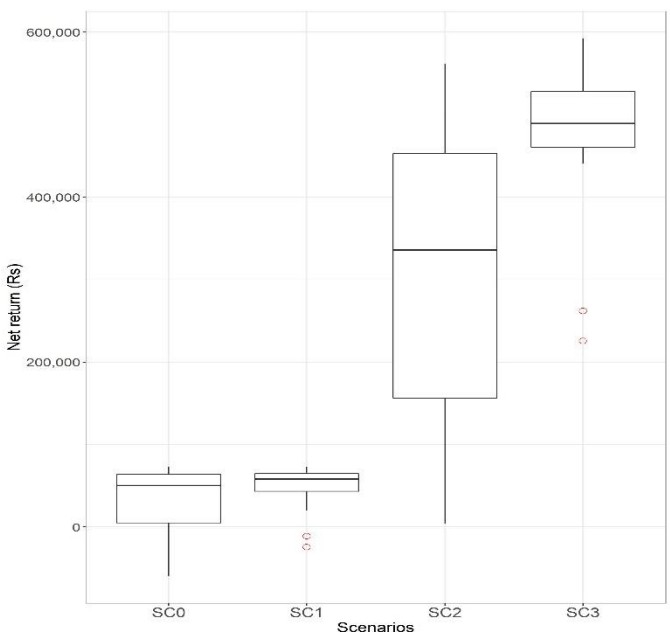

**Figure 4.** Net return distribution over the 15-year simulation period by scenario.

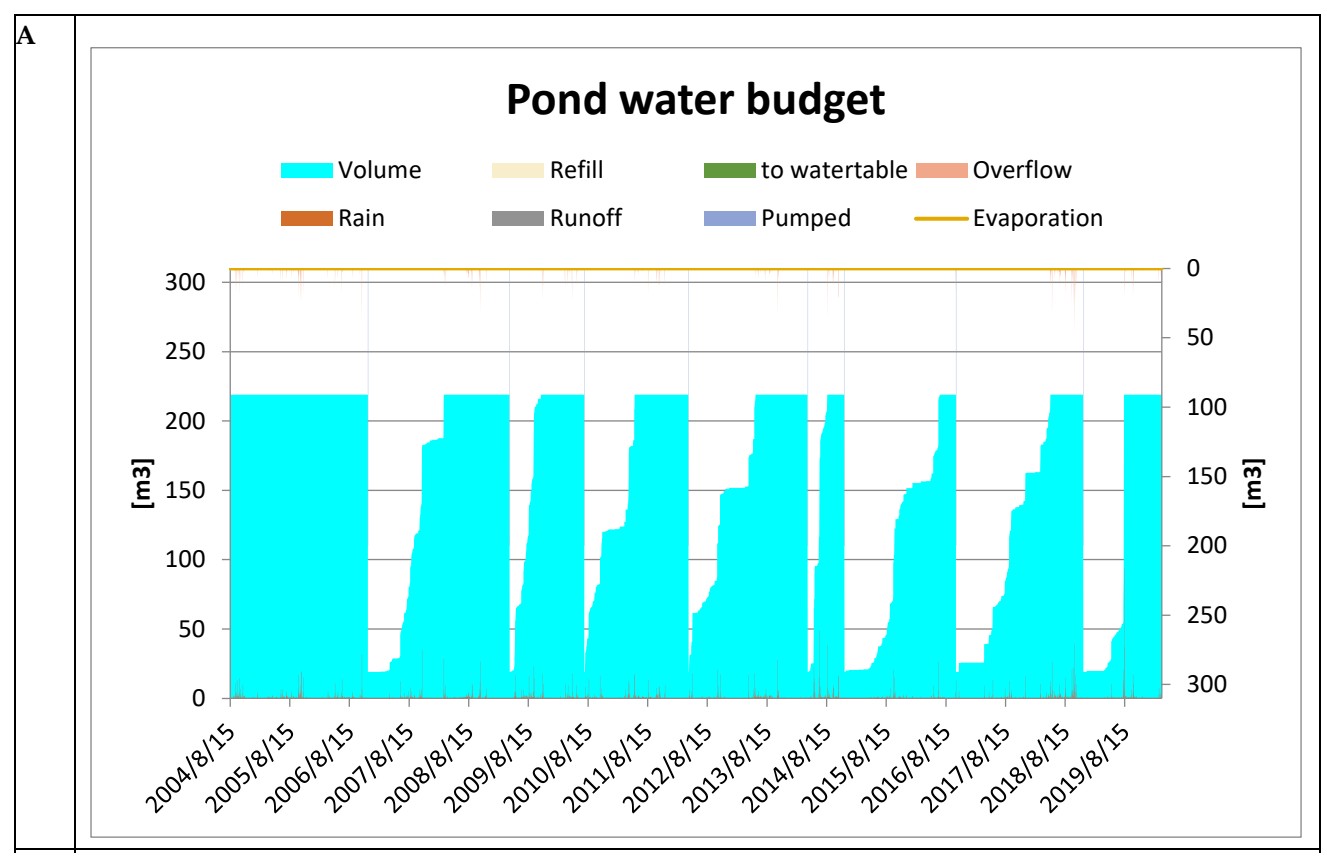

**Figure 5.** *Cont.*

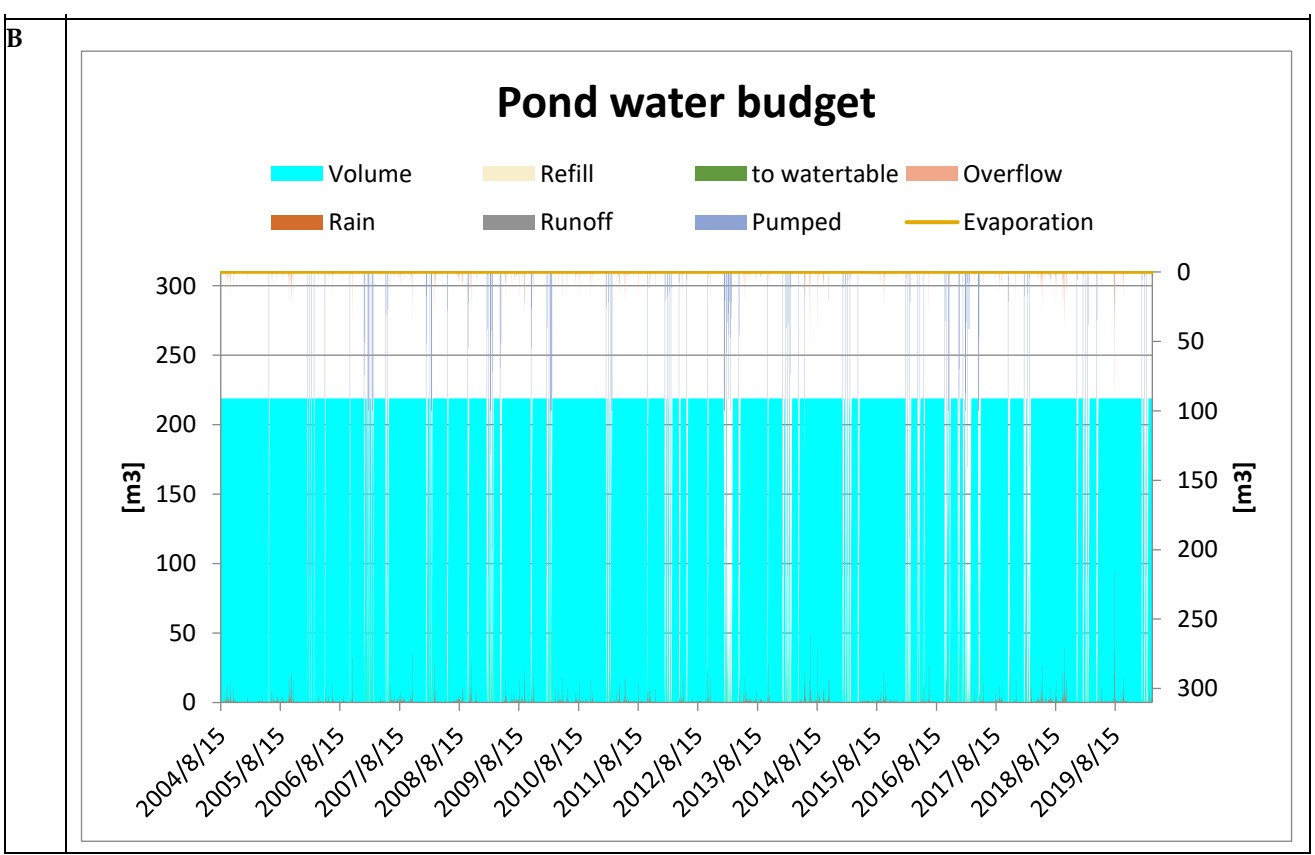

**Figure 5.** Pond water budget for Sc1 (**A**) and Sc3 (**B**).

The last indicator we propose is water table evolution due to the use of the different systems (Figure 6). The use of the pond in Sc1 has a low impact on the water table (comparing Sc0 and Sc1). The slight decrease in water table levels in Sc1 comes from the reduction of crop failures, which consequently reduces the water that is drained into the groundwater, as it is used by the saved crops. The 100% irrigated cropping system of Sc2 impacts the level of the water table, because the long-term pumping pushes the water table to a quasi-constant decline. On the other hand, in Sc3, and in addition to the phenomenon explained before (the reduction of crop failure), the amount of water pumped is important because all the water available from the pump for one day is pumped even if there is no irrigation demand or if the irrigation demand is lower than the water available from the pump.

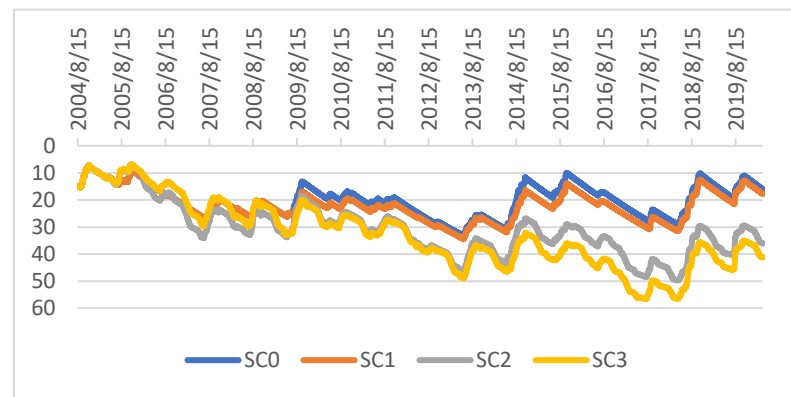

**Figure 6.** Evolution of the water table depth depending on the four scenarios of farming practices.

## 4. Discussion

So far, no model exists to deal with farmers' decision-making to irrigate their farms with multiple irrigation sources in the Indian context. NIRAVARI has been developed to be used as a tool by policy makers and technical advisors to assess different policies better placed to manage water resources in the broader context of climate change. To develop such a tool, we reused simple but robust, already existing models, such as the FAO model for crop growth and other simple models for other processes. The originalities of NIRAVARI regarding other irrigation models in India (see for example) are: (1) the decision-making model and its ability in testing a large range of decisions based on simple criteria; (2) its ability to study the distribution of water from the same source to different crops; (3) its ability to study two irrigation sources simultaneously; and (4) to consider the feedback between the cropping system and the water source.

Instead of using complex modelling languages, we chose a wide-spread language (VBA), allowing for object-oriented modelling and programming. Each element of our farm system is then modeled as an object, and a multiplicity of objects (such as beles or crops) are dealt with as structured containers. The use of Excel allows users to manage simple graphic user interfaces with validation processes in order to avoid any false input to the model parameters.

The scenarios tested show that rainfed systems can maintain a balance in the water table, but the income from these systems is very low and highly variable due to climate variability. However, adding a pond to this system reduces the vulnerability of these farms to climate variability while preserving the water table. On the other hand, a 100% irrigated system with high-added value crops leads to a significant improvement in income, but a drastic decrease in the water table. Indeed, this system leads to an instability of income that systematically decreases with the lack of water. Adding a pond buffer to this system limits the variability of income, but accentuates the decline of the water table, which limits the sustainability of the groundwater resource. These simulations can provide valuable considerations to policymakers, to decide on production systems and water storage technologies, and for their use to be promoted.

Creating such a parsimonious, simple and handy model is also limited to the range of model applications, and the interpretation of some outputs should be considered with care. For example, the choice of the FAO-56 single coefficient formalism [33] implies that the model represents evaporation and transpiration as a single flux, and therefore, calculating the crop water use efficiency for different scenarios is not possible. Similarly, as in the FAO-56 formalism, there is no impact of the water stress on crop water demand, the stress is probably overestimated for rainfed crops—adapted to reduce their demand during drought—compared to strictly irrigated crops. Finally, the impact of water stress on the marketable yield, although drastically simplified (with no account for the disproportionate effect of stress during few critical phenologic stages and being more pronounced for water intensive crops than for more rustic ones), implies that the economic outputs of the model must be considered only as rough estimates.

Moreover, the size of the jeminu, the number and size of plots being fixed, the crop rotation being a forced variable for each simulation, and the adaptation of farmers to variations of water availability can be only partially accounted for in the model. This is the case of the "Jevons paradox" observed in Indian systems, where access to water-saving irrigation technologies can induce an increase of the irrigated area, and therefore, a faster depletion of the aquifer [37].

Finally, while one original feature of NIRAVARI is to account for the feedback between agricultural practices and groundwater resources, it can only represent one farm at a time, and therefore, the results of the simulations should not be interpreted as predictions of what is likely to happen in a farm, which could be surrounded by other farms with very different practices, all foraging the same aquifer. Instead, users should keep in mind that the type of questions to ask the model are rather of the type: "What is likely to happen if all farmers in a small region adopt the same practices as the one that is simulated in NIRAVARI?".

## 5. Conclusions

NIRAVARI was developed to help farmers and advisers, and to provide some analysis to policy-makers, move toward schemes they would like to implement. NIRAVARI can be used for a large range of questions and can, thanks to its simplicity (parsimonious) and genericity, be applied much beyond the Indian context. Crop parametrization, ground water transfer process and farm structure are easily changeable to represent a wide range of water management situations from countries other than India. Even if NIRAVARI was at first aimed at policy-makers, it can also be used as a training media for students to understand the impacts of irrigated agriculture on groundwater resources.

As a follow-up of the initial meeting with the policy-makers during a workshop held in Bangalore in March 2019, NIRAVARI is due to be presented to the officers of the Watershed Department in December 2022.

## 6. Patents

The NIRAVARI model is freely available on request to the main corresponding author.

**Author Contributions:** Software, J.-E.B.; validation, J.-E.B., MB and L.R.; conceptualization and methodology, J.-E.B. and M.B.; writing—original draft preparation, J.-E.B.; formal analysis, J.-E.B., M.B. and L.R.; writing—review and editing, J.-E.B., M.B., L.R. and M.S.; funding acquisition, L.R. and M.S. All authors have read and agreed to the published version of the manuscript.

**Funding:** This study was supported by the ATCHA project [ANR-16-CE03-0006] and the Environmental Research Observatory M-TROPICS (https://mtropics.obs-mip.fr/, accessed on 11 October 2022), which is supported by the University of Toulouse, IRD and CNRS-INSU.

**Institutional Review Board Statement:** Not applicable.

**Informed Consent Statement:** Not applicable.

**Data Availability Statement:** Not applicable.

**Acknowledgments:** The authors wish to thanks the different trainees who worked on the project and the different members of the ATCHA project for their constructive comments on the NIRAVARI model.

**Conflicts of Interest:** The authors declare no conflict of interest.

## Appendix A. Equations of the Biophysical Model

Let us call $b$ the index to represent a bele and $t$ the index for the time.

### Appendix A.1. Soil Water Budget

The state variable of interest is the soil water content ($W(b,t)$, m$^3$). Water inputs ($iW(b,t)$, m$^3 \cdot$d$^{-1}$) are irrigation ($I(b,t)$, m$^3 \cdot$d$^{-1}$) and part of the rainfall in case of runoff ($Rf(b,t)$, m$^3 \cdot$d$^{-1}$). Runoff is based on the curve number formalism [38]. Water outputs ($oW(b,t)$, m$^3 \cdot$d$^{-1}$) are actual evapotranspiration ($E_a(b,t)$, m$^3 \cdot$d$^{-1}$, see crop process) and drainage ($Dr(b,t)$, m$^3 \cdot$d$^{-1}$). Drainage occurs when the soil water capacity is full (tipping bucket formalism, [39]).

$$\begin{cases}
\delta W(b,t) = iW(b,t) - oW(b,t) \\
iW(b,t) = Rf(b,t) + I(b,t) \\
oW(b,t) = E_a(b,t) \\
Rf(b,t) = [R(t) - runoff(t)] * size(b)/1000 \\
\begin{cases}
if(k_c(b,t) > 0) \ then \begin{cases} if\left(\frac{W(b,t-1)}{W_x(b)}\right) > f_w \ then \ E_a(b,t) = min(W(b,t-1), k_c(b,t)\cdot E_0(t)) \\ else \ E_a(b,t) = min\left(W(b,t-1), \left(\frac{W(b,t-1)}{W_x(b)}\right) * f_w * k_c(b,t)\cdot E_0(t)\right) \end{cases} \\
else \ E_a(b,t) = min(W(b,t-1), k_s * E_0(t) * size(b)/1000
\end{cases} \\
if(W(b,t-1) + \delta W(t) > W_x(b)) \ then \begin{cases} Dr(b,t) = (W(b,t-1) + \delta W(t)) - W_x(b) \\ W(t) = W_x(b) \end{cases} \\
else \begin{cases} W(t) = W(t-1) + \delta W(b,t) \\ Dr(b,t) = 0 \end{cases}
\end{cases} \tag{A1}$$

where $k_c$ is the evaporation crop coefficient (see Appendix A.2. Crop Processes), $k_s$ is the evaporation bare soil coefficient, $W_x(b)$ is the maximum soil water capacity of bele $b$ (m$^3$) and $size(b)$ is the size of the bele (m$^2$)

### *Appendix A.2. Crop Processes*

Let us call $A(t)$ the age of the crops (in days). From crop sowing to crop harvest (see 2.3.1. ), $A(t)$ follows a simple linear function: $A(t) = A(t-1) + 1$.

Crop coefficient ($kc(t)$) is modelled by a multilinear function from FAO56 [33] (Figure A1)

$$\begin{cases}
if(0 < A(t) < d_1) \ then \ kc(t) = kc_1 \\
\\
else \ if(d_1 < A(t) < (d_1 + d_2)) \ then \ kc(b,t) = (A(t) - d_1) * \frac{(kc_2 - kc_1)}{d_1} + kc_1 \\
else \ if \ ((d_1 + d_2) < A(t) < (d_1 + d_2 + d_3)) \ then \ kc(b,t) = kc_2 \\
else \ if \ ((d_1 + d_2 + d_3) < age(t) < (d_1 + d_2 + d_3 + d_4)) \ then \ kc(b,t) = ((d_1 + d_2 + d_3 + d_4) - A(t)) * \frac{(kc_2 - kc_3)}{d_4} + kc_3 \\
else \ kc(b,t) = kc_3
\end{cases} \tag{A2}$$

where $d_1$, $d_2$, $d_3$ and $d_4$ are duration between growing phases (in days) and $kc_1$, $kc_2$ and $kc_3$ are specific $kc$ values. All these parameters are crop dependent.

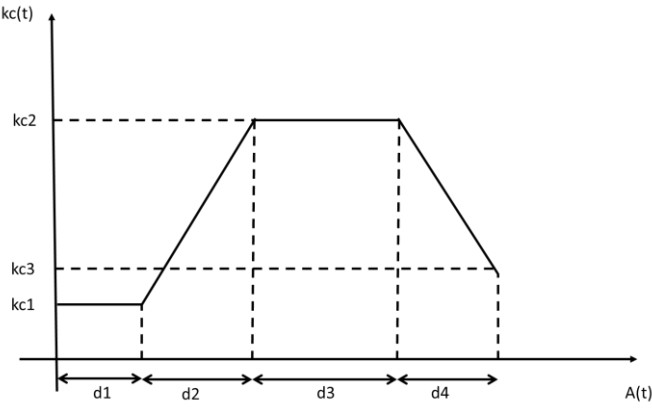

**Figure A1.** The multistage crop coefficient. From FAO56 [33].

Representing Crop Water Stress

The crop water stress is calculated using a simple algorithm. If the crop requires water and no water is provided, its stress increases by one. If the level of stress reaches a given

threshold, then the crop dies. The level of stress can be zeroed if a sufficient amount of water is provided to the crop:

$$\begin{cases} if\left\{\frac{AET(t)}{MET(t)} < p_1 \; and \; I(t) = 0\right\} then \; N_s(t) = N_s(t-1)+1 \\ if(I(t)+R(t) > p_2) \; then \; N_s(t) = 0 \\ if \; N_s(t) \geq p_3 \; then \; crop \; die \end{cases} \tag{A3}$$

where $AET(t)$ is the actual evapotranspiration, $MET(t)$ is the maximum evapotranspiration, $I(t)$ is the amount of irrigation water provided, $N_s(t)$ is the stress level, $R(t)$ is the daily rainfall $p_1$, $p_2$ and $p_3$ are crop dependant parameters. Modifying $p_1$, $p_2$ and $p_3$ allows to represent different crop sensitivity to water stress.

### *Appendix A.3. Pump Process*

The pump flow rate ($f$, m$^3$ s$^{-1}$) follows a power function depending on the depth of the ground watertable (see Appendix A.5. Water Table Processes). This is based on pump flows measurements on the Berambadi watershed [26,32].

$$f(t) = a \cdot H(t-1)^b \tag{A4}$$

where $a$ and $b$ are parameters and $H$ represents the height of the water table (m)

### *Appendix A.4. Pond Processes*

The pond is modelled as a right prism (Figure A2). Pond water contents ($W_p(t)$, m$^3$), depends on the water level in the pond ($h_p(t)$, m). Water input ($iW_p(t)$, m$^3 \cdot$d$^{-1}$) are rainfall ($R_p(t)$, m$^3$), refill by the pump when water is available ($Rf(t)$, m$^3$—see management processes) and runoff ($Ro(t)$, m$^3$). Water output are pumping for irrigation ($I_p(t)$, m$^3$—see management processes), surface evaporation ($Ep(t)$, m$^3$) and recharge of the ground water table if the pond is permeable ($D_p(t)$, m$^3$).

$$\begin{cases} \delta W_p(t) = iW_p(t) - oW_p(t) \\ iW_p(t) = R_p(t) + Rf(t) + Ro(t) \\ oW_p(t) = E_p(t) + I_p(t) + D_p(t) \\ R_p(t) = R(t) * surf_0/1000 \\ \begin{cases} if(R(t) < \gamma_1 then \; Ro(t) = 0 \\ else \; Ro(t) = \gamma_2 \cdot (R(t) - \gamma_1) \end{cases} \\ E_p(t) = \min(W_p(t-1), E_0(t) \cdot surf(t-1)/1000 \\ D_p(t) = \partial \cdot W_p(t-1) \end{cases} \tag{A5}$$

where $\gamma_1$ and $\gamma_2$ are parameters to deal with the runoff process, $surf_0$ is the upper surface area when the pond is full (m$^2$), $surf(t-1)$ is the actual upper surface area (m$^2$). Due the geometrical structure of the pond, there $W_p$ and $surf$ are linked. Detailed calculations are not given here.

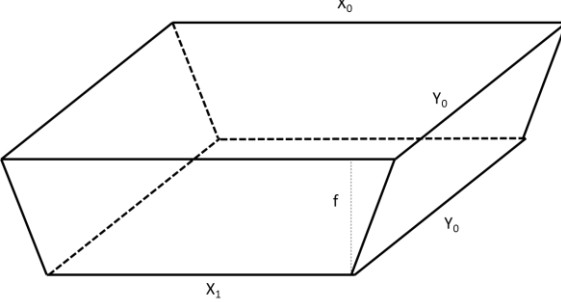

**Figure A2.** The pond geometrical representation.

*Appendix A.5. Water Table Processes*

The ground watertable height ($H(t)$, m) varies depending on the net recharge ($nR(t)$, $\mathrm{m^3 \cdot d^{-1}}$), ie the difference between input (drainage from the different beles and recharge by the pond) and output (irrigation to the different beles and lateral losses ($Q(t)$, $\mathrm{m^3 \cdot d^{-1}}$).

$$
\begin{cases}
H'(t) = H(t-1) + nR(t)/S_y \\
nR(t) = \left(D_p(t) + \sum_{b=1}^{n} D(b,t)\right) - \sum_{b=1}^{n} I(b,t) \\
if\left(H'(t) > H_x\right) \text{ then } H'(t) = H_x \\
Q(t) = (H'(t) - H) \cdot S_y \cdot \alpha' \\
H(t) = H'(t) - Q(t) \, / \, Sy
\end{cases}
\tag{A6}
$$

where $H(t)$ is an intermediate variable, $S_y$ is the specific yield of the aquifer, $\alpha$ is the groundwater recession coefficient.

**Appendix B**

| Jeminu | | |
|---|---|---|
| size | 1 | (ha) |
| coordonates | 0 | (x,y) |
| electricity | 4 | (hours) |
| nbBele | 2 | |
| Soil Type (1. black soil; 2. red soil) | 1 | |
| CN | 85 | |
| catchment extension factor | 1 | [1…] |
| **Pump** | N | Has a pump |
| A | 50.8464 | |
| B | -0.783 | |
| refill | N | Y/N |
| minimum amount to refill | 0 | (m3) |
| Optimize ressource | Y | {0,1} |
| **Pond** | N | Has a pond |
| x0-value | 10 | (m) |
| y0-value | 10 | (m) |
| z0-value | 3 | (m) |
| shape X | 1 | (m/m) |
| initial filling | 1 | [0-1] |
| Use pond | Y | Y/N |
| minimum volume for irrigation | 0 | (m3) |
| minimum use pump to refill | 0 | (m3) |
| percolation coefficient | 0 | [mm/m] |
| Evaporation reduction factor | 0 | |
| Remove pond surface from jeminu surface | N | Y/N |
| **Water table** | | |
| Hmin | 0 | m |
| Hmax | 80 | m |
| Sy | 0.03 | SI |
| Alpha | 0.0005 | SI |
| H0 | 65 | m |
| **Irrigation technic** Add Remove | Furrow | Sprinkler |
| Amount (mm) | 40 | 20 |

| Priorities | | |
|---|---|---|
| Crop priority | 0 | |
| Stress priority | 2 | 1 |
| Crop cycle priority | 3 | 2 |
| Irrigation technic priority | 0 | |
| **Irrigation crop decis** Add Remove | Horse gram | Sorghum |
| Crop type | Horse gram | Sorghum |
| Priority | 0 | 0 |
| Irrigation starting date (jday) | 1 | 1 |
| Irrigation ending date (jday) | 100 | 130 |
| Actual/Maximum potential evaporation threshold | 0.45 | 0.45 |
| **Bele** Add set of practices | Plot1 | Plot2 |
| Bele priority | 1 | 2 |
| AWmax (mm) | 120 | 120 |
| initial AWC (%) | 1 | 1 |
| kc bare soil | 0.15 | 0.15 |
| Wilting point (%) | 0.66 | 0.66 |
| Crop & Irrigation | Sorghum | Sunflower |
| Sowing date | 20/04 | 20/04 |
| Harvesting date | 28/08 | 28/08 |
| Irrigation technic | None | None |
| Crop & Irrigation | Horse gram | Maize |
| Sowing date | 01/09 | 01/09 |
| Harvesting date | 10/12 | 04/01 |
| Irrigation technic | None | None |

**Figure A3.** Parameters of Sc0.

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
