# Peer review of "NIRAVARI: A Parsimonious Bio-Decisional Model for Assessing the Sustainability and Vulnerability of Rainfed or Groundwater-Irrigated Farming Systems in Indian Agriculture"

_water, doi:10.3390/w14203211_

Round 1

Reviewer 1 Report

Dear editor and authors,

Thank you for the opportunity to review the paper entitled ,, Niravari: A parsimonious bio-decisional model for assessing the sustainability and vulnerability of rainfed or groundwater irrigated farming systems in Indian-agriculture,,.

Research was well designed and detailed results were presented. However, some aspects are not very clear, it should be further improved for possible publication.

The comments are:

1. Line 131-134- My concern is that your survey was conducted in 2016, are the results still representative? because crop variations and climatic conditions change from year to year.Also, more information is needed  about the interview (what were the interview questions? what objectives used? where did you get the questions from (references studies used?. These could be presented.

2. I recommend introducing a separate section with the conclusions of the study.

3. and I am interested in your suggested patterns or strategies for improving the crop diversity and agriculture sustainability, but little information could be found.

Best regards.

Author Response

please find in the attached file our responses to your comments.

Reviewer 2 Report

The studied issue is original and brings new interesting knowledge for professional practice in the field of theory and methodology. The visual interpretation and also interpretation of findings is at good level and corresponds to the profile of a scientific journal. Despite this, I have several comments on the manuscript. In the Discussion section I recommend to elaborate the opinions of  other experts on this issue.  In the Discussion section I do not find a citation with number 34. Furthermore, it is necessary to add a Conclusion chapter into the manuscript, which would summarize the processed partial results, the problems that have arisen and new knowledge or it would also bring suggestions for future research.

Author Response

Please find attached our responses to your comments

Cheers

Reviewer 3 Report

Major recommendations

1.     The state of the research field should be reviewed and key publications cited. There are only 23 citations. Among them, 7 references were published over 10 years ago. The analysis is rather weak. There are no statistics. The research gap should be revealed. This section has to be improved.

2.     The purpose of the article is not clear. The authors should mention the main aim of the paper. Otherwise, it is difficult to assess the results and conclusions of the article.

3.     The novelty of the paper is questionable. It is necessary to emphasize the novelty of the study and its main ideas.

Minor recommendations

1.     The Abstract Section: This section must summarize the article’s main findings. In my opinion, the authors should add more details.

2.     Line 72: “[23] is an interesting attempt, …..”. The wording, for example,  “O’Keeffe et al. [23] presented an interesting attempt …..“ is better.

3.     Lines 92-93: “https://mtropics.obs-92 mip.fr/ )” It is better to use a title of website.

4.     Please Explain what the term “jeminu” means.

5.     Line 157: Why did authors used sublevel “Irrigation model at the jeminu level”?

6.     Line 160: “… developed in Bergez et al. (2001).” The authors should use unified links.

7.     As to formulas, the authors should follow the instructions for authors.

8.     Figure 5: There are 8 points in a legend. It is difficult to understand Figure 5. This figure should be improved.

9.     Lines 203-216: The authors should correct these paragraphs.

10. Table 1: The authors should correct the table 1. It is necessary to explain the following: 1/(1+2+3); 2/(1+2+3); 3/(1+2+3).

11. Line 253: What does Sc0 stand for?

12. Some references are too old: 33, 34, and 36.

Author Response

Please find attached our responses to review comments.

Cheers

Reviewer 4 Report

Lines 67,70 and others - please give full names of abbreviations when first used
There is too much information repeated several times, e.g. irrigation methods, types of irrigation equipment, field organisation, . leave one in the best place.
Figure 5 is difficult to read, we do not know if these are daily or weekly budgets, but for clarity maybe just give budgets for 3 growing seasons and 15 years, giving us 45 columns.
How the authors define crop stress. The formula given only shows how it is calculated, but they have not given at what values we are talking about crop stress
The study presented in the paper is, in my opinion, chaotic and contains many information gaps and the results are more than general. In addition, a lot of information is repeated, which should not be the case.
Firstly, a diagram of the irrigated area is not shown, as I understand that one pump serves more than one 'jeminu'. If so, how many. Also, where is the irrigation pond located and how is the irrigation organised in each system, as we only know that either furrow, drip and sprinkler, was it continuous irrigation or intermittent, what water and soil conditions determined the decision to irrigate.
Secondly, the purpose of the article is not stated and the scope of the work and research is not made clear. One can guess that these are the results of research from 2004-2019 and that some kind of model is to be created on the basis of this research. If this is the case, when presenting the research results from the experiments, it would be worthwhile to provide the rainfall levels during the growing seasons in each crop and the water requirements of the crops grown, which would show the reader with what deficiency the crops would have without irrigation. I am not from the same climate zone as the authors and I do not know whether the sowing period is standardised for all 'jeminu' or whether each 'jeminu' sows at its own discretion. It would be worth providing such information as well.
Thirdly, table 2 is completely incomprehensible to me - what is meant by crop failure, what are the parameters of this. On what basis is it concluded that there is failure or not. Instead of such a vague entry, I would suggest providing specific data, i.e. showing the average yield level of the most frequently cultivated crops (3-5) and the number of years with a yield lower than the average by some %, as well as giving the coefficients of variation for average years and for all 'bales'.
Fourthly, the calculation of the net return indicator (NR) is questionable. Firstly, I would expect analyses to be based on yields achieved in the years 204-2019 and not on potential yields, which it is not clear on what basis they were established, and secondly, the 'mean crop water stress' is not, in my opinion, reliable for determining the actual level of yield, since the actual level of yield is determined, in addition to the water stress, by the period in which the water stress occurred, its magnitude in combination with atmospheric conditions, i.e. mainly temperature, sunshine, in addition to soil quality, plant species and cultivar, etc. The net return indicator (NR) is a measure of the actual yield. Adopting this way of calculating the net return indicator will not give meaningful results. I would rather expect the results of the experiments carried out, which can only be the starting point for the development of NIRAVARI.
Fifthly, the reasons for the high proportion of failures in the Sc2 cropping system and nevertheless the high proportion in the Sc3 system in relation to Sk1 have not been explained in the discussion
In conclusion, the absence of the previously stated studies significantly limits the cognitive value of the model and, in my opinion, the tasks implied by the title of the article have not been adequately fulfilled. I suggest the authors to re-write the article from scratch.

Author Response

Please find enclosed our responses to reviewer's comments

Cheers

Round 2

Reviewer 4 Report

The content of the revised article is now much clearer and more understandable.
I am still not satisfied with the explanation of Table 2. It is true that it justifies why 'crop failure' has a similar % failure despite irrigation, but it does not state what has to happen in order to assume that this failure has occurred. What are the parameters of failure. On what basis is it concluded that there is failure or not. Is it a yield lower than the average by any %?

Author Response

Dear reviewer

please find enclosed our proposal to your comment

Sincerely yours
